# Socioeconomic inequalities in childhood and adolescent obesity in Australia: The role of behavioral and biological factors

Nirmal Gautam[1,2,3]*, Aquib Chowdhury[4], Mohammad Mafizur Rahman[2,3], Rasheda Khanam[2,3]

1 Department of Public Health, Faculty of Medical and Allied Science, Karnali College of Health Sciences, Kathmandu, Nepal, 2 School of Business, University of Southern Queensland, Toowoomba, Queensland, Australia, 3 The Centre for Health Research, University of Southern Queensland, Toowoomba, Queensland, Australia, 4 Department of Anaesthetics, Central Queensland Hospital and Health Service, Gladstone, Queensland, Australia

* gnirmal655@gmail.com

## Abstract

### Background

Obesity among children and adolescents is a significant public health concern, influenced by a complex interplay of biological and behavioral factors. However, the extent to which these factors contribute to socioeconomic disparities in obesity remains inadequately understood. Therefore, this study aimed to elucidate the roles of behavioral factors —such as dietary habits, physical activity levels, and outdoor activities—alongside biological factors, including parental body weight, in shaping socioeconomic inequalities in obesity among Australian children and adolescents.

### Methods

This study utilized data from the Birth Cohort (n=5101) and Kindergarten Cohort (n=4983) of the Longitudinal Study of Australian Children (LSAC). LSAC data have been collected biannually since 2004 for the B cohort and since 2000 for the K cohort. The study employed Concentration Index and Decomposition Index analyses to assess the magnitude and to identify the relative contributions of socioeconomic inequalities in obesity, focusing on the contributions of behavioral and biological factors.

### Results

The analysis revealed that socioeconomic disparities in obesity among children and adolescents were significantly influenced by both biological and behavioral factors, as well as household income. Biological factors were found to account for 28.96% of these disparities, while household income contributed 49.17%, and behavioral factors explained 10.36% of the inequalities. Moreover, non-consumption of fatty foods and outdoor activities were found to be associated with a decrease in obesity by $\beta: -0.317$, $\beta: -0.084$, respectively. However, non-consumption of fruits and vegetables and maternal BMI were

**Data availability statement:** Data cannot be shared publicly because data used are confidential. Data are available from the Australian Department of Social Services via the following link: https://dataverse.ada.edu.au/dataverse/lsac.

**Funding:** The author(s) received no specific funding for this work.

**Competing interests:** NO authors have competing interests.

**Abbreviations:** ABS: Australian Bureau of Statistics; AIFS: Australian Institute of Family Studies; AIHW: Australian Institute of Health and Welfare; BMI: Body Mass Index; CI: Concentration Index; DSS: Department of Social Service; OECD: Organization for Economic Cooperation and Development; LSAC: Longitudinal Study of Australian Children; SES: Socioeconomic Status; SDGs: Sustainable Development Goals; UNICEF: United Nations International Children's Emergency Fund; WHO: World Health Organization.

significantly correlated with an increased risk of obesity in children ($\beta$: 0.406) and adolescents ($\beta$: 0.117) respectively.

## Conclusion

These findings provide critical insights into the distribution of obesity across different socioeconomic groups in Australia, highlighting the substantial role of household income and the combined impact of biological and behavioral factors. The results emphasize the importance of developing targeted public health interventions that support families from lower socioeconomic backgrounds to reduce obesity-related disparities.

## Introduction

Obesity is a major public health issue worldwide, affecting over a billion people, including approximately 380 million children and adolescents as of 2020 [1,2]. The incidence of obesity is highest in Asia, impacting socioeconomic and health outcomes significantly [1]. While obesity is a global concern, its distribution and underlying risk factors vary across regions due to differences in socioeconomic conditions, health policies, and lifestyle behaviors [3,4].

Socioeconomic status (SES), which is measured by education, income, occupation, and social class [5], significantly influences obesity rates in children and adolescents [6]. Studies have demonstrated that children from low SES backgrounds tend to have unhealthy eating habits (such as consuming junk food and sugary beverages), physical inactivity, and excessive screen time due to limited access to goods and services [7–9], and significantly associated with obesity. Such environments hinder the adoption of healthy diets and active lifestyles, disproportionately affecting these groups and contributing to social health inequalities [10,11].

In a similar vein, the parental body weight (i.e., parental Body Mass Index (BMI)) recognized as a significant predictor of childhood obesity [12,13]. Studies have shown that having obese parents increases the likelihood of childhood obesity by 28–79% compared to nonobese parents [14,15]. Moreover, studies indicate a significant association between parental BMI and a child's body weight, where children of parents with healthy BMI tend to engage in healthier behaviors, such as regular physical activity and improved dietary habits [16,17]. Conversely, a higher maternal BMI is associated with greater child BMI, increased sedentary behavior, reduced fruit intake, and higher screen time [16]. These findings underscore the critical role of parental body weight status in determining a child's susceptibility to obesity and the importance of early detection and intervention efforts for at-risk children and families [18]. Thus, gaining a comprehensive understanding of the multifaceted factors and distribution of obesity is crucial for addressing obesity inequalities [19].

Given the global burden of obesity and its strong socioeconomic determinants, it is crucial to examine these dynamics in specific national contexts to inform targeted interventions. Australia faces an uneven distribution of obesity, where SES is a stronger predictor of obesity than the country's overall economic level [20,21]. Research by the Australian Institute of Health and Welfare (2020) indicates that 28% of children and adolescents (aged 2–17 years) from low-SES backgrounds are overweight or obese, compared to 21% of their high-SES counterparts, highlighting a disparity in the distribution of resources, opportunities, and goods and services between low- and high-SES households [22,23]. This imbalance contributes to Australia's high burden of obesity, ranking fifth-highest among the Organization for Economic Cooperation and Development (OECD) in 2017–2019 [22,24,25]

Despite the extent of research, examining the distributional effect of biological and behavioral factors on socioeconomic inequalities in obese Australian children and adolescents is relatively rare. This study uniquely integrates behavioral and biological factors with SES to analyze their combined impact on childhood obesity across different socioeconomic groups in Australia, providing new insights into the complex mechanisms underlying the development of obesity. By combining new empirical findings with analyses of the Birth (B) and Kindergarten (K) cohort data from the Longitudinal Study of Australian Children (LSAC), this research offers a comprehensive and novel perspective on the socioeconomic disparities in childhood obesity. While previous studies have examined the relationship between SES and childhood obesity, the simultaneous consideration of biological and behavioral factors within this context is less explored, particularly in the Australian setting [26–30].

Extending the groundwork of previous studies, this study aims to bridge the gaps in the literature through a comprehensive examination of how behavioral and biological factors influence socioeconomic disparities in obesity among Australian children and adolescents. In this study, behavioral factors were defined as the amount of fruits and vegetables, high-fat foods (e.g., chips, cookies, and salty snacks), and sugary drinks (e.g., Coca-Cola or cordial) consumed by the study child, as well as their outdoor time and sedentary activities during their free time indoors [31,32]. External biological factors, on the other hand, are represented by the body mass index (BMI) of the child's parents [33]. This study contributes to the literature by addressing two critical research questions: (i) To what extent do behavioral and biological factors influence the incidence of childhood and adolescent obesity, and (ii) how do these factors contribute to socioeconomic disparities in childhood obesity among high- and low-SES groups? By exploring these research questions, this approach not only fills a significant gap in the existing literature but also provides actionable insights that could inform targeted interventions and policies aimed at reducing health disparities and achieving a broader impact on societal health equity. The findings from this study are pivotal, as they enhance the understanding of how interlinked factors contribute to obesity, paving the way for more effective public health strategies and interventions.

## Materials and methods

### Study setting, study design, and sample

This study used the data from the B cohort and K cohort of the LSAC, which is an ongoing national representative survey commenced in 2003/04 and 1999/00, respectively, and conducted by the Australian Institute of Family Studies (AIFS), the Department of Social Service (DSS), and the Australian Bureau of Statistics (ABS). The LSAC employed a cross-sequential study design, utilizing a multistage cluster sampling method, and collected data biennially, primarily from the biological mothers (P1) in 95% of cases. If the biological mother was unavailable, data were gathered from fathers, grandparents, adoptive parents, or stepparents. Adolescents aged 12 and older provided their data directly to the LSAC. The data collection involved structured questionnaires for both parents and adolescents from B and K cohorts.

In this secondary data analysis study, we examined the relationship between SES, parental BMI, and children's behavioral factors, as well as their distributional impact on childhood and adolescent obesity. Data were collected from P1 for children aged 2–12 years, corresponding to waves 2–6 for the B cohort and waves 1–4 for the K cohort. Additionally, data from adolescents aged 12–13 and 14–16 years were included in waves 5 and 6 for the K cohort. The analysis spans a 10-year follow-up period, encompassing five key data points. The selection of waves 2–7 for the B cohort and waves 1–6 for the K cohort was based on data availability for this study's objectives.

For the K cohort, the initial dataset at wave 1 comprised 4,983 participants during the years 1999/00. Follow-up continued through wave 6, resulting in a final sample of 3,537 participants in the years 2009/10. For the B cohort, the initial dataset at wave 2 included 4,606 participants in the years 2005/06, with follow-ups extending through wave 7, culminating in a final sample of 3,381 participants in the years 2015/16 (Fig 1). The detailed LSAC methodology is available elsewhere [34].

## Dependent/outcome variable

Obesity is the outcome variable of this study and is measured by body mass index (BMI) score. BMI is a universal and validated measurement tool used to categorize humans as underweight, normal weight, or obese based on tissue mass and height [35]. In the LSAC, body weight was measured using Tanita body fat scales, and height was measured using laser stadiometers following standardized protocols [36]. BMI classification in this study follows the age- and sex-specific thresholds defined by Cole et al. (2000, 2007). Cole et al. (2000, 2007) developed age- and sex-specific BMI cut-off points to classify children and adolescents into underweight, normal weight, overweight, and obesity categories. These thresholds are derived based on international growth reference data, aligning with adult BMI cut-offs ($\geq 25\,\text{kg/m}^2$ for overweight and $\geq 30\,\text{kg/m}^2$ for obesity) at age 18. The method uses the 85th percentile to define overweight and the 95th percentile for obesity, ensuring a standardized approach across different populations.

## Independent variables

**Measures of behavioral factors.** A balanced lifestyle is an important predictor of health and well-being [37,38]. Regular physical activity and a nutritious diet improve health and

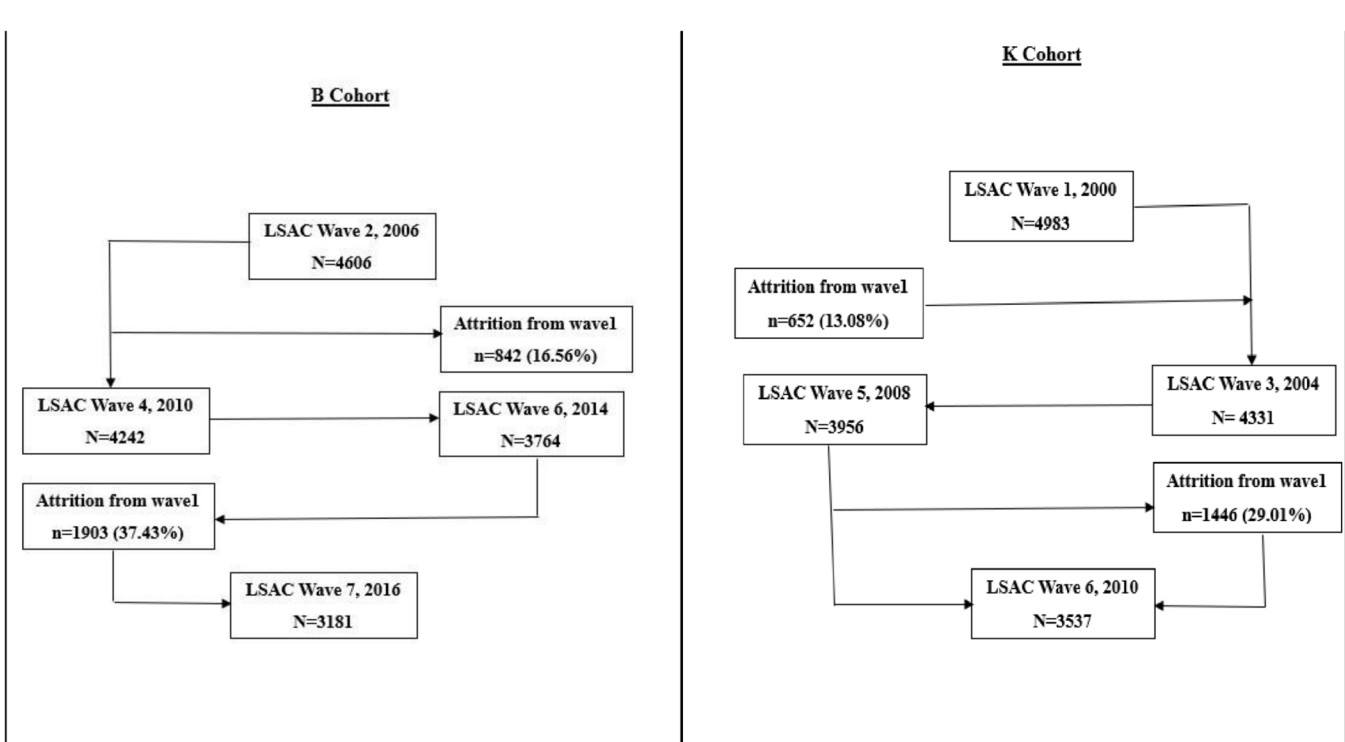

**Fig 1. Participants diagram of the study.**

quality of life but are also indirectly associated with better education, employment, and social opportunities and outcomes [39–41]. Conversely, children who consume unhealthy foods and engage in limited physical and outdoor activities tend to experience negative effects on their social, emotional, and behavioral outcomes [42–45]. Therefore, adopting healthy lifestyle behaviors from an early stage of life is a crucial factor for the health of children and adolescents [46–48]. Based on the previous literature [11], this study used the following five indicators to reflect the behavioral factors among children and adolescents: (i) consumption of fruits and vegetables, (ii) intake of sugary beverages, (iii) consumption of high-fat foods, (iv) activities during free time, and (v) participation in outdoor activities.

## Consumption of fruit and vegetables, fatty food, and sugary beverages

In the LSAC, the consumption of fruits and vegetables was measured by the following question: (i) *within 24 hours, how many times did the study child eat cooked vegetables, raw vegetables, or fresh fruits*? Whereas consumption of fatty food was measured by (ii) *within 24 hours, how many times does the study child eat foods including meat pies, hamburgers, hotdogs, sausages, sausage rolls, French fries, savory snacks, biscuits, doughnuts, and chocolates*? Similarly, the consumption of sugary beverages was measured by (iii) *how many times did the study child drink sweet fruit juices, Coca-Cola, cordial, or lemonade within 24 hours?*

All responses were recorded on an ordinal scale: (i) not at all, (ii) once, or (iii) twice or more than twice a day. We converted these responses into two groups: (1) nonconsumption of fruit/vegetables and/or consumption of fatty food and sugary drinks more than or equal to one time was coded as "0"; (2) consumption of fruit/vegetables and/or nonconsumption of fatty and sugary foods in the last 24 hours was coded as "1".

## Activities during free time

The LSAC collected information on how the study children spent their free time, which revealed that the children engaged in various recreational activities, such as riding bikes, dancing, walking, watching TV, playing video games, and using electronic devices. These activities were categorized by physical activity status as follows: riding bike/dancing/walking (coded as 1) and screen time (watching TV or using electronic devices) (coded as 0).

## Outdoor activities

Moreover, this LSAC assessed outdoor activities by measuring five different levels of activity performed by the study child in a month. These activities included (i) watching sports events with parents and other family members, (ii) swimming with parents and other family members, (iii) attending school or community events with parents and other family members, (iv) visiting the library with parents and other family members, and (v) attending religious services with parents and other family members. All the responses were recorded using a four-point scale, with "not at all" coded as 0, "sometimes" coded as 1, "once in a fortnight" coded as 2, and "very often" coded as 3. Table 1 presents detailed information on the leisure time and outdoor activity items, as well as other behaviors of children and adolescents, including descriptive statistics.

## Measures of biological factors

Biological factors are important determinants of children's health and subsequent physical and psychological growth and development [49–51]. Previous literature has indicated that higher

**Table 1. Descriptive statistics of the variables used in this studyPooled results for the Regression and decomposition of socioeconomic inequalities in obesity for B-cohort (aged 2–12 years).**

| Variables | Pooled B- Cohort | Pooled K- Cohort |
|---|---|---|
| *Control Variables* | **Mean/percent (%)** | |
| Age in years | 0.472 | 0.242 |
| **Gender** | | |
| Male | 0.508 (50.8%) | 0.508 (49.2%) |
| Female | 0.492 (49.2%) | 0.492(49.2%) |
| **Areas of residence** | | |
| Accessible city areas | 0.957 | 0.956 |
| Not accessible regional areas | 0.043 | 0.044 |
| **Dependent Variable** | | |
| Body Mass Index (BMI) of children | 17.21 | 18.11 |
| **Independent Variables** | | |
| **Consumption of fruit and Vegetables (e.g., fresh fruits, cooked vegetables, and raw vegetables) in the last 24 hours** | | |
| One or more than one time in a day | 0.948 | 0.955 |
| Not at all | 0.052 | 0.045 |
| **Consumption of fatty foods (e.g., French fries, savory food, biscuits, and pie) in the last 24 hours** | | |
| one or more than one time in a day | 0.881 | 0.881 |
| Not at all | 0.119 | 0.119 |
| **Drinking sugary beverages (e.g., fruit juice, soft drink/cordial)** | | |
| One or more than one time in a day | 0.677 | 0.743 |
| Not at all | 0.323 | 0.257 |
| **Activities during free time** | | |
| Riding bike/dancing/ walking | 0.703 | 0.747 |
| Screening time | 0.297 | 0.253 |
| **Outdoor activities** | | |
| Gone for swimming with parents or other family members in a month? | 0.77 | 0.68 |
| Involved in a school event or community event with parents or other family members in a month? | 0.462 | 0.494 |
| Watched a sports event with parents or other family members in a month? | 0.607 | 0.682 |
| Attend a religious service with parents or other family members in a month? | 0.278 | 0.314 |
| Visited a library with parents or other family members in a month? | 0.322 | 0.329 |
| **Biological factor** | | |
| Mother BMI | 25.75 | 25.83 |
| Father BMI | 26.65 | 26.99 |
| **Household's income** | | |
| Lowest income (500 AUD or less per week) | 0.382 | 0.45 |
| lowest to medium (501–999 AUD per week) | 0.413 | 0.372 |
| Medium to highest (1000–1999 AUD per week) | 0.177 | 0.156 |
| Highest (more than 2000 AUD per week) | 0.028 | 0.022 |
| **Mother education** | | |
| Postgraduation | 0.09 | 0.077 |
| Undergraduate | 0.283 | 0.249 |
| Certificate/Diploma | 0.61 | 0.657 |

*(Continued)*

**Table 1.** (Continued)

| Variables | Pooled B- Cohort | Pooled K- Cohort |
|---|---|---|
| *Control Variables* | Mean/percent (%) | |
| Year 12 or below | 0.017 | 0.017 |
| **Father education** | | |
| Postgraduation | 0.08 | 0.081 |
| Undergraduate | 0.203 | 0.184 |
| Certificate/Diploma | 0.687 | 0.713 |
| Year 12 or below | 0.03 | 0.022 |
| **Mother employment** | | |
| Full-time employed | 0.411 | 0.353 |
| Part-time Employed | 0.306 | 0.379 |
| Unemployed | 0.283 | 0.268 |
| **Father employment** | | |
| Full-time employed | 0.914 | 0.912 |
| Part-time employed | 0.037 | 0.038 |
| Unemployed | 0.049 | 0.05 |

parental BMIs (i.e., obesity) prior to conception predisposes offspring to higher BMIs at all life stages [52,53]. In this study, paternal and maternal BMI calculated from LSAC measurements (height in meters squared and weight in kilograms) was used as a primary biological factor and analyzed as a continuous variable.

## Measurement of income

Income was aggregated by combining the fathers' and mothers' weekly income from all sources and termed household disposable income. Next, we equivalized this disposable income using the OECD approach outlined by ABS (2006), which allowed us to measure socioeconomic status and construct the income components of the CI. The following formula was used to calculate the household disposable income.

$$\text{Household income} = \frac{\text{Household disposable income}}{1 \times \text{first adult} + 0.5 \times \text{additional adult} + 0.3 \times \text{additional child}}. \quad (1)$$

## Other variables

This study used the education and employment status of the father and mother to control for other socioeconomic status characteristics in the analysis. The analysis also took into account sociodemographic variables (such as age, sex, and place of residence). Table 1 reports the descriptive statistics of these variables.

## Potential bias

Although cohort studies are generally more robust against bias than other observational approaches, such as cross-sectional studies, it remains essential to be vigilant about three specific biases: selection bias, informational bias, and confounder bias [54]. The data collection methods employed by LSAC strictly conform to premier international protocols for longitudinal cohort studies, specifically designed to mitigate biases linked to geographical locations and nonresponses [55,56]. While it is difficult to identify every potential confounder bias, this

study utilized both crude and adjusted regression models to explore the impact of possible confounders, such as demographics, on the relationships between behavioral factors, biological factors, socioeconomic status, and obesity.

## Ethical approval and consent to participate

The Longitudinal Study of Australian Children (Growing Up in Australia) has received ethical approval from the Ethics Committee of the Australian Institute of Family Studies. The authors were granted permission to use the LSAC data, referenced under number 412282. They accessed the dataset titled 'Growing Up in Australia: Longitudinal Study of Australian Children (LSAC) Release 9.0 C2 (Waves 1-9C)' on April 28, 2023. Hence, the datasets that were analyzed or created during this study are bound by the confidentiality agreement that was signed. However, interested parties must comply with certain restrictions and sign confidentiality agreements. To request access to the data, individuals should reach out to the Australian Department of Social Services via the following link: https://dataverse.ada.edu.au/dataverseuser.xhtml?editMode=CREATE

## Statistical analysis

Initially, we used descriptive statistics, including frequency, mean, and standard deviation, to summarize the variables in our study. Second, we analyzed longitudinal data using Ordinary Least Squares (OLS) regression, accounting for repeated observations using a random effects model. This approach allowed us to assess elasticities and explore the dynamics between behavioral factors, biological factors, and obesity in children and adolescents. We also employed a concentration index to evaluate socioeconomic disparities in obesity among youths. Additionally, decomposition analysis was performed to measure the contribution of each factor to socioeconomic inequalities in children and adolescents with obesity. Moreover, we handled instances of missing data using a straightforward imputation technique. All statistical procedures were carried out using R

## Concentration index (CI)

The concentration index (CI) was used to measure socioeconomic inequalities. The CI is a widely used method to measure and compare the degree of socioeconomic inequality. Positive and negative signs indicate the distribution of health variables. A positive CI indicates that the relationship between living in a standard and health is distributed toward a pro-rich distribution, while a negative CI indicates a poor-poor distribution of health. In addition, the CI is bounded by -1–1. The CI is determined by calculating the ordinary least squares (OLS) estimate of β [57,58].

$$2\sigma_r^2\left(\frac{H_i}{\bar{H}}\right) = \alpha + \beta r_i + \varepsilon_i \qquad (2)$$

where $\alpha$ is the intercept, $\beta$ represents the measure of the relationship between each predictor variable $x$ and obesity, $H_i$ is the health variable of interest, $i^{th}$ is the sample population, $\bar{H}$ is a mean of the health variables of interest, $\sigma_r^2$ is the variance of fractional rank, and $r_i = \frac{1}{N}$ is the fractional rank of the $i^{th}$ study population rank by income or other measure of socioeconomic status (SES), ordered from the least to the most economically advantaged distribution (i.e., $i=1$ for poorest and $i=N$ richest).

## Decomposition analysis

Decomposition analysis was used to assess the factors contributing to the socioeconomic inequalities associated with obesity. This study used the Wagstaff et al. (2003) approach to examine how much inequality in a particular variable contributes to socioeconomic inequality in obesity among children and adolescents. Wagstaff et al. [59] reported that if health is a linear function of factors (k), such as demographics, lifestyle, and socioeconomic status (SES), then the CI represents a weighted sum of the socioeconomic inequalities found within these factors. Hence, the following regression model decomposes the CI:

$$H_i = \alpha + \sum_i^k \beta_k x_{ik} + \varepsilon_i \tag{3}$$

where $\alpha$ is the intercept, $\beta$ represents a parameter that quantifies the association between each explanatory factor x and obesity, and $\varepsilon$ represents the error term. According to Wagstaff et al. [59], $CI$ of $H_i$ can be decomposed into the contributions of factors that explain obesity in children and adolescents as follows:

$$CI = \sum_j \eta_j CI_j + GC_u / \bar{x} \tag{4}$$

where $\eta_j$ is the elasticity and is calculated using $\eta_j = \frac{\beta_i \bar{h_i}}{\bar{H}}$, and $CI_j$ is the concentration index of each predictor. According to Eq. 4, the product of the elasticity of each factor and the CI gives the contribution of the factor to the inequalities, and $GC_u$ represents the generalized concentration index for the error term. The percent contribution is calculated by $(C_j / CI) \times 100 = percent\ contribution\ of\ CI$. To calculate the decomposition of the concentration index, we followed these steps:

1. We conducted a panel data multiple linear regression analysis from equation (2) that accounted for various factors that could affect the variable of interest, including biological and behavioral characteristics, age and gender of children, parents' education and employment, and household income.

2. Afterwards, we calculated the mean value of the variables of interest and multiplied the mean value of each variable by its corresponding coefficient from the regression model, yielding a measure of elasticity. This measure indicated how sensitive the variable of interest (outcome variable) was to changes in each of the explanatory variables.

3. Next, we used the R library (rineq) package to calculate the concentration index of each variable.

4. Finally, we multiplied the concentration index by the elasticity of each explanatory variable to determine its contribution to overall inequality in the distribution of obesity in children and adolescents. We applied the same procedure to a pooled regression model that combined data from multiple periods. The details of the methodology can also be found elsewhere [60]

## Results

### Descriptive statistics of the variables

Table 1 shows the pooled descriptive statistics of the variables used in this study for the B-cohort (Wave 2 to Wave 7) and K-cohort (Wave 1 to Wave 6) cohorts. The study revealed

that the average BMI of children was 17.21 for the B-cohort and 18.11 for the K-cohort. Additionally, the average paternal BMIs were 26.65 and 26.99 for the B and K cohorts, respectively, and the average maternal BMIs were 25.75 and 25.83 for the B and K cohorts, respectively (Table 1). According to Table 1, the average BMI of parents in the LSAC sample is greater than the normal range as defined by the WHO for adults aged over 20 years, which is 18.5 to 24.9. On the other hand, the average values of consumption of fruits and vegetables, as well as intake of fatty foods, were similar in both cohorts (Table 1). Compared to the B-cohort, the older K-cohort on average consumed more sugary beverages than did the B-cohort (K-cohort: 0.743, B-cohort: 0.677) but spent less time on screen activities (K-cohort: 0.253, B-cohort: 0.297).

Table 1 demonstrates that the parents of B-cohort children are comparatively wealthier and more educated than the parents of K-cohort children. Moreover, parents of B-cohort children are more likely to be employed full-time, whereas the unemployment rate and part-time employment of fathers are almost identical for both cohorts. Finally, 95% of the study children came from accessible urban areas.

S3 Appendix presents an intriguing finding: the consumption of fruits and vegetables, physical activity, and outdoor activity gradually decreased over time (waves 7 and 6 in the B-cohort and K-cohort, respectively). As a result, this pattern could progressively shift the BMI distribution toward overweight or obese Australian youth.

Table 2 presents the regression results (Panel A), decomposition analysis (Panel B), and contributing factors by percentage (Panel C) for cohort B. In the regression results, behavioral and biological factors were found to be statistically significant and have the expected signs (Table 2: Panel A). The results show that nonconsumption of fruits and vegetables was significantly correlated with an increase in obesity $\beta: 0.406$. Panel A further shows that nonconsumption of fatty foods decreases the probability of being obese $\beta: -0.317$. Similarly, an increase in physical and outdoor activities was found to be associated with a decrease in obesity by $\beta: -0.684$, and $\beta: -0.084$, respectively. These results were consistent with our expectations. This study revealed that, on average, for each one-point increase in the BMI of either the father or mother, the obesity rate in their children increased by $\beta: 0.117$, and $\beta: 0.076$, respectively (Table 2: Panel A). This result suggested that obesity in children can be predicted in large part by their parents' body weight.

Table 2 (Panel A) shows that parental income has a significant effect on the relationship between SES and childhood obesity. We found that children from the lowest income quintile were significantly more likely to be obese or overweight than their counterparts were. Interestingly, the mother's education has no effect on BMI, whereas the father's education does. Specifically, children whose fathers had a certificate or diploma were more likely to have a higher BMI. Maternal part-time employment was associated with higher BMI in children, whereas the father's employment status had no effect on BMI.

Furthermore, the first row of each variable in Panel B presents the elasticity ($\eta$), which indicates the expected direction (positive or negative) of the change in the outcome variable (i.e., obesity) in response to a one-unit change in the independent variable. Similarly, the second row of each variable represents the concentration index ($CI$), which represents the degree of socioeconomic inequality within the distribution of obesity. A positive (negative) sign of ($CI$) implies that the distribution of obesity skews toward advantaged or wealthier (disadvantaged or poorer) socioeconomic status. The third row of each variable represents the estimated contribution ($Con$) of the examined independent variable to the socioeconomic inequality observed in obese individuals and is calculated as the multiplicative form of elasticity and ($CI$) of each variable divided by the estimated ($CI$).

**Table 2. Regression and decomposition of socioeconomic inequalities in obesity for B-cohort (aged 2 -12 years).**

| Variables | Panel A: Regression results | Panel B: Decomposition analysis | | Panel C: Contributing factors by percentage (%) |
|---|---|---|---|---|
| | Co-eff (SE) | | | |
| **Control variables** | | | | |
| Age **in years** | 0.324 | $\eta$ | 0.139 | -0.34 |
| | (0.006) | CI | 0.000 | |
| | *** | Con | 0.0001 | |
| Female (=1) (*Ref: Male*) | −0.052 | $\eta$ | −0.001 | 1.22 |
| | (0.038) | CI | 0.011 | |
| | | Con | 0.0000 | |
| *Areas of residence (Ref: accessible city areas)* | | | | |
| Not accessible regional areas (=1) | 0.072 | $\eta$ | 0.000 | -1.49 |
| | (0.094) | CI | 0.105 | |
| | | Con | 0.0000 | |
| **Behavioral factors** | | | | |
| Consumption of fruit and vegetables: not at all (=1) (*Ref: one or more than one time in a day*) | 0.406 | $\eta$ | 0.022 | -2.79 |
| | (0.093) | CI | 0.001 | |
| | *** | Con | 0.0000 | |
| Consumption of fatty food: not at all (=1) (*Ref: one or more than one time in a day*) | −0.317 | $\eta$ | −0.016 | 1.85 |
| | (0.064) | CI | 0.001 | |
| | *** | Con | 0.0000 | |
| Drinking sugary beverages: not at all (=1) (*Ref: one or more than one time in a day*) | 0.052 | $\eta$ | 0.002 | **0.36** |
| | (0.045) | CI | −0.002 | |
| | | Con | 0.0000 | |
| Activities in free time: riding bike/dancing/walking (=1) (*Ref: screen time*) | −0.684 | $\eta$ | −0.028 | 8.04 |
| | (0.044) | CI | 0.003 | |
| | *** | Con | −0.0001 | |
| Outdoor activities | −0.084 | $\eta$ | −0.013 | 10.01 |
| | (0.017) | CI | 0.007 | |
| | *** | Con | −0.0001 | |
| **Biological factors** | | | | |
| Mother or maternal BMI | 0.117 | $\eta$ | 0.175 | 47.46 |
| | (0.004) | CI | −0.002 | |
| | *** | Con | -0.0004 | |
| Father or paternal BMI | 0.076 | $\eta$ | 0.118 | -4.71 |
| | (0.006) | CI | 0.000 | |
| | *** | Con | 0.0000 | |
| **Household Income (Ref: lowest)** | | | | |
| lowest to medium (=1) | −0.190 | $\eta$ | −0.005 | **33.28** |
| | (0.049) | CI | 0.064 | |
| | *** | Con | −0.0003 | |
| Medium to highest (=1) | −0.095 | $\eta$ | -0.001 | **8.93** |
| | (0.066) | CI | 0.079 | |
| | | Con | -0.0001 | |

*(Continued)*

**Table 2.** (Continued)

| Variables | Panel A: Regression results | Panel B: Decomposition analysis | | Panel C: Contributing factors by percentage (%) |
|---|---|---|---|---|
| | Co-eff (SE) | | | |
| Highest (=1) | −0.095 | $\eta$ | 0.000 | **1.15** |
| | (0.131) | CI | 0.065 | |
| | | Con | 0.0000 | |
| **Mother's education (Ref: Postgraduation)** | | | | |
| Undergraduate (=1) | −0.093 | $\eta$ | −0.002 | 7.36 |
| | (0.079) | CI | 0.042 | |
| | | Con | −0.0001 | |
| Certificate/Diploma (=1) | 0.094 | $\eta$ | 0.003 | 11.41 |
| | (0.079) | CI | −0.030 | |
| | | Con | −0.0001 | |
| Year 12 or below (=1) | 0.039 | $\eta$ | 0.000 | 0.13 |
| | (0.170) | CI | −0.029 | |
| | | Con | 0.0000 | |
| **Father's education (Ref: Postgraduation)** | | | | |
| Undergraduate (=1) | 0.086 | $\eta$ | 0.001 | -6.79 |
| | (0.086) | CI | 0.058 | |
| | | Con | 0.0001 | |
| Certificate/Diploma (=1) | 0.232 | $\eta$ | 0.009 | 27.09 |
| | (0.082) | CI | −0.026 | |
| | ** | Con | −0.0002 | |
| Year 12 or below (=1) | 0.382 | $\eta$ | 0.001 | 0.31 |
| | (0.140) | CI | −0.004 | |
| | ** | Con | 0.0000 | |
| **Mother's employment (Ref: Full-time)** | | | | |
| Part-time Employed (=1) | 0.329 | $\eta$ | 0.006 | -18.18 |
| | (0.050) | CI | 0.027 | |
| | *** | Con | 0.0002 | |
| Unemployed (=1) | −0.194 | $\eta$ | −0.003 | -19.99 |
| | (0.051) | CI | −0.055 | |
| | *** | Con | 0.0002 | |
| **Father's employment (Ref: Full-time)** | | | | |
| Part-time Employed (=1) | 0.034 | $\eta$ | 0.000 | 0.38 |
| | (0.107) | CI | −0.045 | |
| | | Con | 0.0000 | |
| Unemployed (=1) | −0.061 | $\eta$ | 0.000 | -2.06 |
| | (0.096) | CI | −0.104 | |
| | | Con | 0.0000 | |
| CI of BMI | – | – | **−0.001** | 100 |
| Total estimated contribution | – | – | **−0.00096** | 98.2810814 |

Standard deviations (SD) are on parenthesis; (2) '*', '**' and '***' indicate statistical significance at 10%, 5% and 1% level. Lowest income = 500 AUD or less per week; lowest to medium (501–999 AUD per week), medium to highest (1000–1999 AUD), and highest income (more than 2000),: $\eta$ = elasticity; CI= concentration Index; con =contribution of the variable to socio-economic inequality.

In Table 2 (Panel B), the nonconsumption of sugary beverages, higher maternal BMI, and lower paternal educational attainment (e.g., certificate/diploma/year 12 or below) were more concentrated among children from families with SES disadvantages. On the other hand, nonconsumptions of fatty foods, riding bikes/dancing/walking in free time, and participation in outdoor activities were significantly more common among those from advantaged SESs. Hence, these variables demonstrate a distinct socioeconomic inequality in the population distribution of obese individuals.

Table 2 (Panel C) presents the contribution factors of each independent variable by percentage. The results showed that the consumption of sugary beverages (036%), fatty foods (1.85%), activities during free time (8.04%), participation in outdoor activities (10.01%), maternal BMI (47.46%), low to medium income (33.28%), medium to high income (8.93%), highest parental income (1.15%), and paternal education at the certificate/diploma level (27.09%) were the key factors contributing to the observed socioeconomic inequalities of childhood obesity. These findings emphasize the need to address the underlying socioeconomic factors that underpin childhood obesity.

Moreover, the range of socioeconomic inequalities was found to be between -0.002 and -0.009, with an average of -0.001 (S1 Appendix). This finding suggests that childhood obesity is disproportionately concentrated among socioeconomically disadvantaged groups. Furthermore, the analysis revealed that household income, biological factors, and behavioral factors contributed to socioeconomic inequalities in children with obesity by 43.4%, 42.8%, and 17.5%, respectively (Fig 2, B cohort).

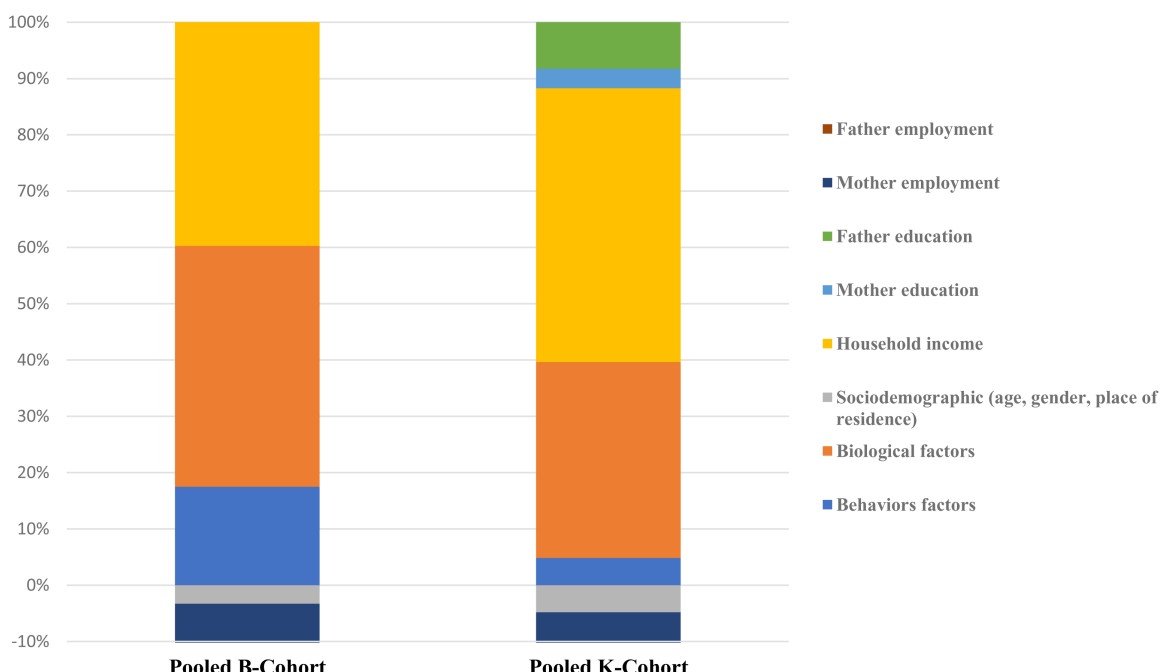

**Fig 2. Contribution of key variables to socioeconomic inequalities on obesity by cohort.**

**Table 3. Regression and decomposition of socioeconomic inequalities in obesity for K-Cohort (aged 4 - 16 years).**

| Variables | Panel A: Regression Results | Panel B: Decomposition analysis | | Panel C: Contributing factors by percentage (%) |
|---|---|---|---|---|
| **Control variables** | Co-eff (SE) | | | |
| Age in years | 0.530 | $\eta$ | 0.273 | -1.20 |
| | (0.007) | CI | 0.002 | |
| | *** | Con | 0.0005 | |
| Female (=1) (*Ref: Male*) | 0.003 | $\eta$ | 0.000 | 0.04 |
| | (0.042) | CI | -0.005 | |
| | | Con | 0.0000 | |
| **Areas of residence (Ref: accessible city areas)** | | | | |
| Not accessible regional areas (=1) | -0.096 | $\eta$ | 0.000 | -0.34 |
| | (0.104) | CI | -0.015 | |
| | | Con | 0.0000 | |
| **Behavioral factors** | | | | |
| Consumption of fruit and vegetables: not at all (=1) (*Ref: Ref: one or more than one time in a day*) | 0.060 | $\eta$ | 0.001 | -0.17 |
| | (0.109) | CI | 0.001 | |
| | | Con | 0.0000 | |
| Consumption of fatty food: not at all (=1) (*Ref: Ref: one or more than one time in a day*) | -0.094 | $\eta$ | -0.011 | -0.33 |
| | (0.069) | CI | 0.000 | |
| | | Con | 0.0000 | |
| Drinking sugary beverages: not at all (=1) (*Ref: Ref: one or more than one time in a day*) | 0.028 | $\eta$ | 0.005 | 3.10 |
| | (0.052) | CI | -0.007 | |
| | | Con | 0.0000 | |
| Activity during free time: Riding bike/dancing/walking (=1) (*Ref: screening time*) | -0.271 | $\eta$ | -0.016 | 2.88 |
| | (0.052) | CI | 0.002 | |
| | *** | Con | 0.0000 | |
| Outdoor activities | -0.143 | $\eta$ | 0.005 | -0.63 |
| | (0.019) | CI | 0.001 | |
| | *** | Con | 0.0000 | |
| **Biological factors** | | | | |
| Mother or maternal BMI | 0.155 | $\eta$ | 0.202 | 23.05 |
| | (0.004) | CI | -0.001 | |
| | *** | Con | -0.0002 | |
| Father or paternal BMI | 0.075 | $\eta$ | 0.098 | 11.45 |
| | (0.005) | CI | -0.001 | |
| | *** | Con | -0.0001 | |
| **Household Income (Ref: lowest** | | | | |
| lowest to medium (=1) | 0.056 | $\eta$ | -0.005 | 38.50 |
| | (0.053) | CI | 0.069 | |
| | | Con | -0.0004 | |
| Medium to highest (=1) | 0.365 | $\eta$ | -0.002 | 10.42 |
| | (0.075) | CI | 0.050 | |
| | *** | Con | -0.0001 | |

*(Continued)*

**Table 3.** (Continued)

| Variables | Panel A: Regression Results | Panel B: Decomposition analysis | | Panel C: Contributing factors by percentage (%) |
|---|---|---|---|---|
| **Control variables** | Co-eff (SE) | | | |
| Highest (=1) | 0.866 | $\eta$ | 0.000 | -0.29 |
| | (0.163) | CI | 0.017 | |
| | *** | Con | 0.0000 | |
| **Mother's education (Ref: Postgraduation)** | | | | |
| Undergraduate (=1) | -0.048 | $\eta$ | 0.000 | 0.61 |
| | (0.094) | CI | 0.022 | |
| | | Con | 0.0000 | |
| Certificate/Diploma (=1) | -0.006 | $\eta$ | 0.002 | 2.60 |
| | (0.093) | CI | −0.015 | |
| | | Con | 0.0000 | |
| Year 12 or below (=1) | -0.060 | $\eta$ | 0.000 | 0.27 |
| | (0.192) | CI | 0.044 | |
| | | Con | 0.0000 | |
| **Father's education (Ref: Postgraduation)** | | | | |
| Undergraduate (=1) | 0.121 | $\eta$ | 0.001 | -4.18 |
| | 0.096 | CI | 0.034 | |
| | | Con | 0.0000 | |
| Certificate/Diploma (=1) | 0.475 | $\eta$ | 0.017 | 25.46 |
| | 0.091 | CI | −0.014 | |
| | *** | Con | −0.0002 | |
| Year 12 or below (=1) | 0.018 | $\eta$ | 0.000 | 0.13 |
| | 0.173 | CI | -0.015 | |
| | | Con | 0.0000 | |
| **Mother's employment (Ref: Full-time)** | | | | |
| Part-time Employed (=1) | -0.353 | $\eta$ | −0.005 | 2.64 |
| | 0.054 | CI | 0.005 | |
| | *** | Con | 0.0000 | |
| Unemployed (=1) | -0.499 | $\eta$ | −0.003 | -11.81 |
| | 0.062 | CI | −0.042 | |
| | *** | Con | 0.0001 | |
| **Father's employment (Ref: Full-time)** | | | | |
| Part-time Employed (=1) | 0.031 | $\eta$ | 0.000 | -0.68 |
| | 0.118 | CI | −0.045 | |
| | | Con | 0.0000 | |
| Unemployed (=1) | 0.283 | $\eta$ | 0.000 | 2.50 |
| | 0.105 | CI | −0.092 | |
| | ** | Con | 0.0000 | |
| CI of BMI | – | – | 0.0001 | 100 |
| Total estimated contribution | – | – | −0.0010 | 99.99 |

standard deviations (SD) are on parenthesis; (2) '*', '**' and '***'' indicate statistical significance at 10%, 5% and 1% level. Lowest income = 500 AUD or less per week; lowest to medium (501–999 AUD per week), medium to highest (1000–1999 AUD), and highest income (more than 2000),: $\eta$ = elasticity; CI= concentration Index; con =contribution of the variable to socio-economic inequality.

## Regression and decomposition of socioeconomic inequalities in obesity for K-Cohort (aged 4–16 years)

Table 3 summarizes the regression results (Panel A), decomposition analysis (Panel B), and contributing factors to socioeconomic obesity inequalities by percentage (Panel C). Table 3 (Panel A) shows that behavioral factors (only Activity during free time and outdoor activity), biological factors, household income (except lowest income), paternal educational attainment (only certificate/diploma), and maternal employment had statistically significant associations with childhood obesity. The results suggest that outdoor activities significantly reduce the probability of being obese by $\beta: -0.143$ in children and adolescents. Subsequently, an increase in nonsedentary activities such as riding bikes/dancing/walking during recreational time was found to be associated with a decrease in obesity $\beta: -0.271$. On the other hand, the results suggest that an increase in one unit of paternal BMI or maternal BMI increased the risk of childhood obesity by $\beta: 0.075$ or $\beta: 0.155$, respectively.

Household income (e.g., middle highest and highest income) and lower paternal educational attainment (certificate/diploma) were positively associated with obesity. This implies that a decrease of one unit in household income (e.g., highest income) and paternal education (certificate/diploma) significantly increases obesity by $\beta: 0.866$ and $\beta: 0.475$, respectively. Interestingly, children of unemployed or part-time employed mothers are less likely to be obese than are children of full-time employed mothers. This suggests that unemployed or part-time employed mothers may have more time to monitor and contribute to their children's diet, exercise, and outdoor activities.

Panel B and Panel C of Table 3 illustrate the decomposition analysis and contributing factors, respectively, by percentage. Panel B reveals that outdoor activities, physical activities during recreational time, and household income, are concentrated in pro-rich SES groups. On the other hand, paternal BMI, maternal BMI, lower paternal educational attainment (certificate/diploma), and parental unemployment are concentrated in disadvantaged SES groups. Overall, the skewed distribution of these factors reinforces inherent socioeconomic inequality. Panel C presents the strength with which these factors were associated with obesity by percentage. Physical activity (2.88%), maternal BMI (23.05%), paternal BMI (11.45%), medium to highest income (10.42%), paternal education (certificate/diploma) (25.46%), and maternal part-time employment (2.64%) were the factors most strongly associated with obesity in our sample.

Moreover, S2 Appendix revealed that biological, behavioral, and household income were the main drivers of the unequal socioeconomic distribution of childhood obesity, ranging from -0.0002 to -0.003, 0.000 to 0.0002, and -0.001 to -0.003, respectively, during the survey periods. This indicates that obesity was concentrated among children with SES disadvantages. Overall, when including a richer set of variables, we found robust evidence of household income (48.63%), biological factors (34.8%), and behavioral factors (4.85%), which significantly contribute to socioeconomic inequalities in childhood and adolescent obesity (Fig 2, K cohort).

## Discussion

The impact of socioeconomic inequality on public health issues such as obesity is acknowledged globally, and reducing disparities to prevent obesity is a significant policy objective for every country [61]. Despite this, evidence shows that childhood obesity is pervasive and is increasing globally. This may be due to the challenge of effectively treating established obesity [62–64]. Thus, prevention is a critical strategy to address the burden of obesity by reducing

exposure to common risk factors [33,65]. It is therefore important to recognize these risk factors and gain a better understanding of how they interact with obesity.

Therefore, the first objective of this study was to investigate the effect of biological and behavioral factors in addition to parental socioeconomic status on obesity among children and adolescents using a 12-year follow-up study from the Longitudinal Study of Australian Children. Biological and behavioral factors have significant impacts on childhood obesity in Australia. The findings of this study revealed that children and adolescents from high-SES households had greater exposure to physical activities during their leisure time, such as playing games and cycling, than did those from low-SES households. This difference may be due to parents with high SES having a better understanding of the importance of exercise and healthy lifestyles, leading them to prioritize exercising in their free time and regularly encouraging their children to engage in physical activities. This finding is consistent with previous literature that has linked higher levels of physical activity with a significant reduction in childhood obesity [66].

The study also revealed a correlation between outdoor activities and obesity, even after adjusting for parental SES. Children and adolescents from high SES backgrounds had a greater likelihood of engaging in outdoor activities, which significantly reduced their risk of obesity. This might be attributed to parents with high SES having better social connections, including family members, peers, and teachers, all of whom have direct and indirect effects on children's health and well-being. Conversely, parents with low SES may have limited access to education, lack social cohesion, and have discordant relationships, leading to a lack of awareness of health priorities and behaviors. As a result, they and their children were less likely to participate in outdoor activities and more likely to engage with media and other forms of entertainment, which increased the risk of obesity [67–69].

This study revealed a significant correlation between childhood and adolescent obesity and maternal obesity, likely due to an altered fetal hypothalamus response to leptin, appetite regulation, and pancreatic beta resulting from maternal obesity. These results indicate that maternal obesity increases offspring body weight. These findings align with other literature that suggests a greater risk of obesity in children and adolescents if their parents are obese [51,70,71].

This study revealed that higher parental SES is associated with a lower risk of childhood and adolescent obesity. It is suggested that parents with high income and education are more likely to provide healthy and nutritious food options for their children, leading to healthier lifestyle behaviors and a lower risk of obesity. Similar findings have been reported in other literature, highlighting the role of parental SES background in childhood and adolescent weight status. Additionally, maternal unemployment or part-time employment has been found to be a protective factor against obesity in children and adolescents, as it allows for more opportunities for mothers to supervise their children's habits and make changes to their diet and physical activity. These findings align with previous longitudinal studies, which suggest that increased availability of parents at home reduces the risk of sedentary behavior and obesity [72–75].

The second aim of this study was to examine the distributional impact of biological and behavioral factors on socioeconomic inequities in the obesity status of Australian children and adolescents. The findings indicate that biological factors, behavioral factors, and household income are all significant contributors to the socioeconomic inequalities in obesity among children and adolescents in both cohorts. For instance, parents with lower SES and higher BMI are more likely to have children with higher BMI, as indicated by the positive elasticity of biological factors. Furthermore, a negative CI for biological factors indicates that obesity is more common among children from low SES backgrounds. On the other hand, a positive CI for physical and outdoor activities suggests that these activities are more common among

children from higher SES backgrounds. As a result, children from low-SES backgrounds are less physically active and less involved in outdoor activities, which contributes to obesity and socioeconomic inequalities in individuals with obesity. These findings are consistent with those for the other variables studied.

In summary, this study extends previous research on the link between socioeconomic factors and childhood obesity by identifying the specific behavioral mechanisms through which SES influences obesity outcomes. Utilizing a longitudinal design and decomposition analysis, it provides a dynamic perspective on how SES disparities in obesity persist and evolve over time. Notably, it is one of the first studies to leverage a comprehensive 12-year follow-up dataset to examine the interplay between biological, behavioral, and socioeconomic factors in shaping obesity disparities among Australian children and adolescents. This long-term approach strengthens the findings by capturing obesity risk across different developmental stages. Furthermore, the study employs concentration and decomposition indices to quantify socioeconomic inequalities in obesity, a relatively underexplored area in Australian childhood obesity research. Decomposition analysis, in particular, serves as a crucial methodological tool for understanding how SES influences obesity risk, informing targeted interventions, and supporting evidence-based policymaking.

## Strengths and limitations

To our knowledge, this study effectively used a large contemporary national longitudinal dataset from Australia to investigate the link between behavioral and biological factors and socioeconomic disparities in obesity among children and adolescents, it is not without limitations. First, the issue of unbalanced longitudinal data, characterized by varying observation counts across different time points, could affect the reliability of the conclusions. Second, the occurrence of missing data points for key study variables across multiple waves could influence the findings. Third, there is a potential risk of bias, as respondents may not remember past events or details accurately. Finally, parental BMI is included as a biological factor in this analysis; however, it is inherently intertwined with behavioral factors such as dietary patterns, physical activity, and family lifestyle. Given that parental BMI influences childhood obesity not only through genetic inheritance but also via shared household behaviors, its role as a purely biological determinant should be interpreted with caution. Although to address these confounding effects, we conducted a mediation analysis to examine the mediating effects of parental BMI on child BMI (S4 Appendix). Despite this limitation, this study provides valuable insights into the factors that contribute to obesity among young people in Australia and the importance of addressing socioeconomic inequalities in promoting healthy behaviors and preventing obesity.

## Conclusion

This study provides robust information on childhood and adolescent obesity, revealing that family income and behavioral and biological factors are significant predictors of childhood and adolescent obesity in Australia. Parents of disadvantaged SES backgrounds are less likely to comply with dietary requirements and nutritional recommendations, fostering an observed unequal socioeconomic distribution in obesity among their offspring. Therefore, childhood obesity should be closely monitored with the implementation of multifactorial programs (social, physical, and environmental dimensions) and policies (improving the environment, reducing inequalities, empowering people, and promoting traditional activities and family role model approaches), particularly for disadvantaged, deprived, and underserved populations, in an equitable way.

## Supporting Information

**S1 Appendix. Contribution of key variables by concentration index (B-Cohort).**
(DOCX)

**S2 Appendix. Contribution of key variables by concentration index (K-Cohort).**
(DOCX)

**S3 Appendix. Descriptive statistics for B cohort and K cohort.**
(DOCX)

**S4 Appendix. Mediation analysis of independent variable on the outcome variable (Obesity).**
(DOCX)

## Acknowledgments

The authors express gratitude to the Australian Institute of Family Studies for granting us access to their data. Furthermore, we extend our sincere gratitude to the School of Business at the University of Southern Queensland in Toowoomba, Australia, for providing the perfect atmosphere that greatly contributed to the successful completion of this study.

## Author contributions

**Conceptualization:** Nirmal Gautam, Rasheda Khanam.

**Data curation:** Nirmal Gautam.

**Formal analysis:** Nirmal Gautam.

**Methodology:** Nirmal Gautam.

**Software:** Nirmal Gautam.

**Supervision:** Mohammad Mafizur Rahman, Rasheda Khanam.

**Validation:** Mohammad Mafizur Rahman, Rasheda Khanam.

**Visualization:** Nirmal Gautam.

**Writing – original draft:** Nirmal Gautam.

**Writing – review & editing:** Aquib Chowdhury, Mohammad Mafizur Rahman, Rasheda Khanam.

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
