## [Decision Letter · Decision Letter 0]

23 Jan 2025

PONE-D-24-34935Socioeconomic Inequalities in Childhood and Adolescent Obesity in Australia: The role of behavioral and biological factorsPLOS ONE

Dear Dr. Gautam,

Thank you for submitting your manuscript to PLOS ONE. After careful consideration, we feel that it has merit but does not fully meet PLOS ONE’s publication criteria as it currently stands. Therefore, we invite you to submit a revised version of the manuscript that addresses the points raised during the review process.

We look forward to receiving your revised manuscript.

Kind regards,

Haris Khurram

Academic Editor

PLOS ONE

Journal Requirements:

Additional Editor Comments:

Check the typos and grammatical errors more carefully.

Reviewers' comments:

Reviewer's Responses to Questions

**Comments to the Author**

1. Is the manuscript technically sound, and do the data support the conclusions?

Reviewer #1: Yes

Reviewer #2: Yes

2. Has the statistical analysis been performed appropriately and rigorously? 

Reviewer #1: Yes

Reviewer #2: Yes

3. Have the authors made all data underlying the findings in their manuscript fully available?

Reviewer #1: Yes

Reviewer #2: No

4. Is the manuscript presented in an intelligible fashion and written in standard English?

Reviewer #1: Yes

Reviewer #2: Yes

5. Review Comments to the Author

Reviewer #1: Title:

Socioeconomic Inequalities in Childhood and Adolescent Obesity in Australia: The role of behavioral and biological factors

I have reviewed the manuscript thoroughly. It has good scientific processes. I want to congratulate the authors on their meticulous job. This is a well-conducted and written study. The authors used very clear methods, language and terminology. The concepts and ideas are nicely communicated. However, I suggest the following changes be made prior to publication.

Abstract

1. L21: Replace it with “Obesity among children and adolescents.”

2. L21: Replace the term “interplay.”

3. L29: Start by adding, “This decomposition analysis study utilized data from...” to provide the reader with an idea of the type, function, or method employed in the study.

4. Add the total number of participants included from the Birth Cohort and Kindergarten Cohort of the LSAC.

5. Mention the original time lapses and duration of this longitudinal study.

6. L32: Add “and to identify the relative contributions.”

7. Include additional numerical values from the test results to enhance the results section.

8. Ensure consistent terminology throughout the abstract, such as “biological factors” and “parental body weight,” which are currently mixed up.

Introduction

9. Highlight the originality of this research clearly, providing readers with an understanding of the new knowledge obtained from this study or clarifying whether it is an analysis of two published datasets.

10. L53: Add a reference.

11. L98: Replace the word “wealth.”

12. The transition from general global statistics to specific Australian data is abrupt. Include a clearer linkage and clarify why Australia was chosen as the focus area.

13. Although the section is comprehensive, it attempts to cover too much ground, making it less focused. Three pages are excessive. Details such as dietary examples seem extra; consider making it more concise.

14. Explain the BMI score classification by Cole et al. concisely, including the percentile used in this method.

15. L172: Provide a reference for literature.

16. Conclude with a stronger statement about the implications of this research.

Study design

17. Concisely define the timeline for each wave of data collection for both cohorts, including start and end years for each wave, to ensure readers understand the temporal structure of the study.

18. Reference the reader to previous studies but also include sufficient details here. Explain the Birth Cohort (B) in the same way as you did the Kindergarten Cohort (K).

19. Explain why specific waves (e.g., waves 1–4 for the K cohort and waves 2–6 for the B cohort) were selected for the analysis. Were these choices based on data availability, age range, or other considerations?

20. Include a flow chart explaining the process.

21. Clarify whether the behavioral factors were a part of the LSAC.

22. L235: Specify whether permission was obtained for using the data or if it is publicly available and does not require prior permission.

23. L253: This section is nicely explained.

24. For exclusion criteria, mention if non-English studies were excluded.

Results:

25. Modify Table 1 to start with control variable information, such as age in years, and include percentages by gender (apply this to other tables as well).

26. Nicely explained Table 2, its clear. Under the table note, explain abbreviations and the significance levels denoted by * and ** (apply this to other tables as well).

Discussion:

27. While it is important to align findings with existing literature, highlight the new insights provided by this study that were not addressed before, as the discussion currently lacks clear rationale.

28. L473: The connection here is unclear.

29. Move the limitations section to the end of the discussion and present it as a separate section.

30. If there are recommendations for future research, include them.

Good Luck

Reviewer #2: Please, see the attached file for more details.

The article makes a significant contribution to understanding the interplay between socioeconomic factors and childhood obesity. However, following are two minor observations need to be addressed before its acceptance for publications:

1.

While behavioral factors are included, the study could further explore external environmental influences, such as urban design and access to recreational facilities, which might shape physical activity levels.

2.

Despite the longitudinal design, some interpretations appear static, potentially overlooking temporal shifts in socioeconomic conditions.

6. PLOS authors have the option to publish the peer review history of their article (what does this mean? ). If published, this will include your full peer review and any attached files.

**Do you want your identity to be public for this peer review?** For information about this choice, including consent withdrawal, please see our Privacy Policy .

Reviewer #1: **Yes: ** Syed Ghufran Hadier

Reviewer #2: No

---

## [Author Response · Author response to Decision Letter 1]

18 Feb 2025

18th February 2025

Dr. Haris Khurram,

Editor, PLOS ONE

RE: Manuscript ID: PONE-D-24-34935

Dear Dr. Khurram,

We sincerely appreciate the opportunity to revise and resubmit our manuscript titled "Socioeconomic Inequalities in Childhood and Adolescent Obesity in Australia: The role of behavioral and biological factors" (Manuscript ID: PONE-D-24-34935) for consideration in PLOS ONE. We extend our gratitude to the reviewers for their thoughtful and constructive feedback, which has significantly contributed to refining our study. We have carefully addressed all comments and made necessary revisions to improve the clarity, precision, and impact of our work.

In response to Reviewers comments, we have implemented the following key revisions:

1. Abstract and Introduction: We have revised the abstract to enhance clarity, ensure consistency in terminology, and include additional numerical values from test results. In the introduction, we have explicitly highlighted the originality of our study, improved the transition from global statistics to Australian-specific data, and concisely stated the BMI classification method.

2. Study Design and Methodology: We have refined the explanation of the timeline for each wave of data collection, ensured consistency in describing the Birth and Kindergarten cohorts, and clarified the rationale behind selecting specific waves. Additionally, we have included a study participant flowchart to visually represent the selection process.

3. Results and Discussion: We have modified Table 1 to start with control variables, included percentage distributions for gender, and clarified significance levels in all tables. In the discussion, we have explicitly highlighted novel insights from our study and revised certain paragraphs for better coherence and clarity.

4. Limitations and Future Research: The limitations section has been moved to the end of the discussion, presented as a separate section, and expanded to include additional considerations. We have also included recommendations for future research, emphasizing the exploration of environmental influences on childhood obesity.

We are thankful for the opportunity to revise and resubmit the paper. We hope that the revised version meets your publication standards. We look forward to your decision in due course.

Yours faithfully,

Nirmal Gautam

Corresponding author

University of Southern Queensland

Toowoomba; Queensland; 4350; Australia Ph: 61 7 4631 1256; Fax: 61 7 4631 5597

Email: gnirmal655@gmail.com/nirmal.gautam@usq.edu.au

Web: www.usq.edu.au

Response to Referee #1

Referee’s comment

Abstract

Line 34: L21: Replace it with “Obesity among children and adolescents.”.

Our Response

Thank you for your suggestion. We have now revised it as suggested (Please see L. No 21 in the revised clean manuscript)

Referee’s comment

L21: Replace the term “interplay.”.

Our Response

Thank you for your suggestion. But we humbly disagree to replace the term “interplay”. The term "interplay" suggest a dynamic relationship between elements. So, we keep this as it is.

Referee’s comment

L29: Start by adding, “This decomposition analysis study utilized data from...” to provide the reader with an idea of the type, function, or method employed in the study.

Our Response

We appreciate the reviewer’s suggestion to revise the sentence as “This decomposition analysis study utilized data from...” to provide clarity regarding the study’s methodology. However, we respectfully disagree with this revision for the following reason:” The original wording maintains a logical flow by first introducing the data source and then specifying the analytical techniques used. The reviewer's suggestion could disrupt this clarity by shifting the focus prematurely to decomposition analysis, potentially making it less clear to readers unfamiliar with the method.”

Referee’s comment

Add the total number of participants included from the Birth Cohort and Kindergarten Cohort of the LSAC.

Our Response

Thank you for your suggestion. We have now revised it as suggested

Referee’s comment

Mention the original time lapses and duration of this longitudinal study.

Our Response

Thank you very much for your feedback. We have revised it in the revised manuscript (please see the Abstract: L. No. 31 in the revised clean manuscript).

Referee’s comment

L32: Add “and to identify the relative contributions.”

Our Response

Thank you for your suggestion. We have now revised it as suggested.

Referee’s comment

Include additional numerical values from the test results to enhance the results section.

Our Response

Thank you for your feedback. We have added numerical values from the test results to enhance the results section. (please see the Abstract section: L. No. 40 to 43 in the revised clean manuscript).

Referee’s comment

Ensure consistent terminology throughout the abstract, such as “biological factors” and “parental body weight,” which are currently mixed up.

Our Response

Thank you for your feedback. We have reviewed the abstract and ensured that biological factors” and “parental body weight,” is used consistently throughout.

Introduction

Referee’s comment on the Introduction:

Highlight the originality of this research clearly, providing readers with an understanding of the new knowledge obtained from this study or clarifying whether it is an analysis of two published datasets.

Our Response

Thank you for your comments and for emphasizing the importance of clarifying the originality and contributions of our research. To address the request, we have revised the introduction parts as

“This study uniquely integrates behavioral and biological factors with SES to analyze their combined impact on childhood obesity across different socioeconomic groups in Australia, providing new insights into the complex mechanisms underlying the development of obesity. By combining new empirical findings with analyses of the Birth (B) and Kindergarten (K) cohort data from the Longitudinal Study of Australian Children (LSAC), this research offers a comprehensive and novel perspective on the socioeconomic disparities in childhood obesity. While previous studies have examined the relationship between SES and childhood obesity, the simultaneous consideration of biological and behavioral factors within this context is less explored, particularly in the Australian setting [26-30].” (please see the Introduction section: Lo. No 101 to 109 in the clean revised manuscript).

Referee’s comment

L53: Add a reference.

Our Response

Thank you for your comment. We have now added the appropriate reference.

Referee’s comment

L98: Replace the word “wealth.”.

Our Response

Thank you for your suggestion. We have revised the term “wealth” to “extend of research”

Referee’s comment

The transition from general global statistics to specific Australian data is abrupt. Include a clearer linkage and clarify why Australia was chosen as the focus area.

Our Response

Thank you for the suggestion. We have revised the transition to provide a clearer linkage between the global statistics and the Australian context as follow;

“Given the global burden of obesity and its strong socioeconomic determinants, it is crucial to examine these dynamics in specific national contexts to inform targeted interventions. Australia faces an uneven distribution of obesity, where SES is a stronger predictor of obesity than the country's overall economic level [20, 21]. Research by the Australian Institute of Health and Welfare (2020) indicates that 28% of children and adolescents (aged 2 to 17 years) from low-SES backgrounds are overweight or obese, compared to 21% of their high-SES counterparts, highlighting a disparity in the distribution of resources, opportunities, and goods and services between low- and high-SES households [22, 23]. This imbalance contributes to Australia's high burden of obesity, ranking fifth highest among OECD countries in 2017–2019 [22, 24, 25]” (please see the Introduction section: Lo. No 89 to 98 in the clean revised manuscript).

Referee’s comment

Although the section is comprehensive, it attempts to cover too much ground, making it less focused. Three pages are excessive. Details such as dietary examples seem extra; consider making it more concise.

Our Response

Thank you for your observation. We have now concise the entire introduction to make it clearer and consistent.

Referee’s comment

Explain the BMI score classification by Cole et al. concisely, including the percentile used in this method.

Our Response

Thank you for your feedback. We have now explained the BMI score classification by Cole et al. concisely, including the percentile used in this method. (please see in the method section 158 to 165 in the clean revised manuscript)

“BMI classification in this study follows the age- and sex-specific thresholds defined by Cole et al. (2000, 2007). Cole et al. (2000, 2007) developed age- and sex-specific BMI cut-off points to classify children and adolescents into underweight, normal weight, overweight, and obesity categories. These thresholds are derived based on international growth reference data, aligning with adult BMI cut-offs (≥25 kg/m² for overweight and ≥30 kg/m² for obesity) at age 18. The method uses the 85th percentile to define overweight and the 95th percentile for obesity, ensuring a standardized approach across different populations.”

Referee’s comment

L172: Provide a reference for literature.

Our Response

Thank you for your feedback. We have provided a reference for literature.

Referee’s comment

Conclude with a stronger statement about the implications of this research.

Our Response

Thank you for your feedback. We have concluded with a stronger statement about the implications of this research (please see the Introduction section: Lo. No 121 to 126 in the clean revised manuscript).

“This approach not only fills a significant gap in the existing literature but also provides actionable insights that could inform targeted interventions and policies aimed at reducing health disparities and achieving a broader impact on societal health equity. The findings from this study are pivotal, as they enhance the understanding of how interlinked factors contribute to obesity, paving the way for more effective public health strategies and interventions”

Study design

Referee’s comment

Concisely define the timeline for each wave of data collection for both cohorts, including start and end years for each wave, to ensure readers understand the temporal structure of the study.

Our Response

Thank you for your suggestion. We have concisely defined the timeline for each wave of data collection for both cohorts, including start and end years for each wave as follow:

“This study used the data from the B cohort and K cohort of the LSAC, which is an ongoing national representative survey commenced in 2003/04 and 1999/00, respectively, and conducted by the Australian Institute of Family Studies (AIFS), the Department of Social Service (DSS), and the Australian Bureau of Statistics (ABS). The LSAC employed a cross-sequential study design, utilizing a multistage cluster sampling method, and collected data biennially, primarily from the biological mothers (P1) in 95% of cases. If the biological mother was unavailable, data were gathered from fathers, grandparents, adoptive parents, or stepparents. Adolescents aged 12 and older provided their data directly to the LSAC. The data collection involved structured questionnaires for both parents and adolescents.

In this secondary data analysis study, we examined the relationship between SES, parental BMI, and children's behavioral factors, as well as their distributional impact on childhood and adolescent obesity. Data were collected from P1 for children aged 2 to 12 years, corresponding to waves 2 to 6 for the B cohort and waves 1 to 4 for the K cohort. Additionally, data from adolescents aged 12-13 and 14-16 years were included during waves 5 and 6 for the K cohort. The analysis spans a 10-year follow-up period, encompassing five key data points.

For the K cohort, the initial dataset at wave 1 comprised 4,982 participants during the years 1999/00. Follow-up continued through wave 6, resulting in a final sample of 3,537 participants in the years 2009/10. For the B cohort, the initial dataset at wave 2 included 4,605 participants in the years 2005/06, with follow-ups extending through wave 7, culminating in a final sample of 3,381 participants in the years 2015/16. The detailed LSAC methodology is available elsewhere [32]. ”

Referee’s comment

Reference the reader to previous studies but also include sufficient details here. Explain the Birth Cohort (B) in the same way as you did the Kindergarten Cohort (K).

Thank you for your feedback. I have explained the Birth Cohort (B) in the same way as you the Kindergarten Cohort (K) as follow:

“This study used the data from the B cohort and K cohort of the LSAC, which is an ongoing national representative survey commenced in 2003/04 and 1999/00, respectively, and conducted by the Australian Institute of Family Studies (AIFS), the Department of Social Service (DSS), and the Australian Bureau of Statistics (ABS). The LSAC employed a cross-sequential study design, utilizing a multistage cluster sampling method, and collected data biennially, primarily from the biological mothers (P1) in 95% of cases. If the biological mother was unavailable, data were gathered from fathers, grandparents, adoptive parents, or stepparents. Adolescents aged 12 and older provided their data directly to the LSAC. The data collection involved structured questionnaires for both parents and adolescents from B and K cohorts.

In this secondary data analysis study, we examined the relationship between SES, parental BMI, and children's behavioral factors, as well as their distributional impact on childhood and adolescent obesity. Data were collected from P1 for children aged 2 to 12 years, corresponding to waves 2 to 6 for the B cohort and waves 1 to 4 for the K cohort. Additionally, data from adolescents aged 12-13 and 14-16 years were included in waves 5 and 6 for the K cohort. The analysis spans a 10-year follow-up period, encompassing five key data points. The selection of waves 2–7 for the B cohort and waves 1–6 for the K cohort was based on data availability for this study's objectives.

For the K cohort, the initial dataset at wave 1 comprised 4,983 participants during the years 1999/00. Follow-up continued through wave 6, resulting in a final sample of 3,537 participants in the years 2009/10. For the B cohort, the initial dataset at wave 2 included 4,606 participants in the years 2005/06, with follow-ups extending through wave 7, culminating in a final sample of 3,381 participants in the years 2015/16 (Figure 1). The detailed LSAC methodology is available elsewhere [34]. ”

Referee’s comment

Explain why specific waves (e.g., waves 1–4 for the K cohort and waves 2–6 for the B cohort) were selected for the analysis. Were these choices based on data availability, age range, or other considerations?

Our Response

Thank you for the suggestion regarding the data used in this study. Yes, we have chosen the selective waves 2 to 7 for the B cohort and 1 to 6 waves for the K cohort. These choices are based on the availability of data. We have revised it in the revised manuscript (please see the Method section: Lo. No 144 to 146 in the clean revised manuscript).

Referee’s comment

Include a flow chart explaining the process.

Our Response

Thank you for your suggestions. We have included a study participants flow chart to explain the process. (please see Figure 1).

Referee’s comment

Clarify whether the behavioral factors were a part of the LSAC.

Our Response

Thank you for your comment. Yes, the behavioral factors examined in this study were obtained from the LSAC dataset.

Referee’s comment

L235: Specify whether permission was obtained for using the data or if it is publicly available and does not require prior permission.

Our Response

Thank you for your concern. We have updated the data availability in the revised manuscript as:

“interested parties must comply with certain restrictions and sign confidentiality agreements. To

---

## [Decision Letter · Decision Letter 1]

12 Mar 2025

Socioeconomic Inequalities in Childhood and Adolescent Obesity in Australia: The role of behavioral and biological factors

PONE-D-24-34935R1

Dear Dr. Gautam,

We’re pleased to inform you that your manuscript has been judged scientifically suitable for publication and will be formally accepted for publication once it meets all outstanding technical requirements.

Kind regards,

Haris Khurram

Academic Editor

PLOS ONE

Additional Editor Comments (optional):

Reviewers' comments:

Reviewer's Responses to Questions

**Comments to the Author**

1. If the authors have adequately addressed your comments raised in a previous round of review and you feel that this manuscript is now acceptable for publication, you may indicate that here to bypass the “Comments to the Author” section, enter your conflict of interest statement in the “Confidential to Editor” section, and submit your "Accept" recommendation.

Reviewer #1: All comments have been addressed

Reviewer #2: All comments have been addressed

2. Is the manuscript technically sound, and do the data support the conclusions?

Reviewer #1: Yes

Reviewer #2: Yes

3. Has the statistical analysis been performed appropriately and rigorously? 

Reviewer #1: Yes

Reviewer #2: Yes

4. Have the authors made all data underlying the findings in their manuscript fully available?

Reviewer #1: Yes

Reviewer #2: Yes

5. Is the manuscript presented in an intelligible fashion and written in standard English?

Reviewer #1: Yes

Reviewer #2: Yes

6. Review Comments to the Author

Reviewer #1: All the comments have been thoroughly addressed. The authors have made significant improvements, and the revisions have enhanced the clarity and quality of the article. I am satisfied with its current form and recommend it for acceptance.

Reviewer #2: The minor observation has been addressed and the article is fit for acceptance. Please the attached report.

7. PLOS authors have the option to publish the peer review history of their article (what does this mean? ). If published, this will include your full peer review and any attached files.

**Do you want your identity to be public for this peer review?** For information about this choice, including consent withdrawal, please see our Privacy Policy .

Reviewer #1: **Yes: ** Syed Ghufran Hadier, Shanxi University, Taiyuan 030006, China

Reviewer #2: **Yes: ** Muhammad Aslam

---

## [Editor Report · Acceptance letter]

PONE-D-24-34935R1

PLOS ONE

Dear Dr. Gautam,

I'm pleased to inform you that your manuscript has been deemed suitable for publication in PLOS ONE. Congratulations! Your manuscript is now being handed over to our production team.

Kind regards,

on behalf of

Dr Haris Khurram

Academic Editor

PLOS ONE